# Smallholder Farming during COVID-19: A Systematic Review Concerning Impacts, Adaptations, Barriers, Policy, and Planning for Future Pandemics

Alexander R. Marsden *, Kerstin K. Zander and Jonatan A. Lassa

College of Indigenous Futures, Arts and Education, Charles Darwin University, Darwin, NT 0909, Australia
* Correspondence: s338356@Students.cdu.edu.au; Tel.: +61-439088075

**Abstract:** Our broad aim was to systematically analyse research on the effect of COVID-19 on smallholder farming during 2019–2021 and to discuss how the research could be beneficial to smallholder farm resilience to future pandemics. The review methods were based on PRISMA guidelines, and 53 articles were included in the final review. The review aims to document the social-economic impacts on different groups, barriers and opportunities of smallholder farmers adapting to COVID, and policy options. Barriers to adaptations were considered in only 15% of journal articles, suggesting a research gap. This review highlights the fact that, among others, technology access to ensure information and crisis communication that specifically targets smallholders, as well as multi-layered diversification, serves as good predictors of smallholder adaptation to COVID-19. Multi-layered diversification includes product diversification, market diversification and income stream diversification. This confirms the established knowledge in disasters and livelihood studies where diversification of livelihoods portfolio serves as the key factor to resilience against shocks and crisis. Finally, we summarised the different policy implications arising from the literature. This implies that governments must develop an effective policy-mix that leaves no smallholder farmers behind in future pandemics.

**Keywords:** logistics; supply chains; extension services

## 1. Introduction

COVID-19, a zoonotic disease, has caused unapparelled global disruption, sickness, and death. The threatening nature of the virus has been an urgent concern for governments. Governments sought to limit the spread of COVID-19 and its variants [1,2] and prevent illness and death [3,4] by implementing severe lockdowns and social distancing laws [5–9]. Governments and communities faced unprecedented global and national economic impacts [10]. Food security was a significant concern during the pandemic for governments. Agricultural services were required to respond to many countries' food shortages caused by COVID-19 [11]. However, governments were not always able to respond to farmers' needs because of the crippling effect of COVID-19 on the economy [12]. Furthermore, in many countries, small-scale farming is a vital food source for growers and those who purchase food at their local markets [13].

The travel restrictions introduced to minimise the spread of COVID-19 increased the impacts on and disruptions to farmers [10]. Many small-scale farmers were severely impacted as they could not obtain the necessary inputs and overcome transport interruptions. In many countries, small-scale farmers urgently required support to ensure food supply [7] due to the impact of COVID-19 on supply chains and logistics [10,12]. An analysis by Asegie et al. [13] in Ethiopia found that 88.89% of smallholder farmers' households were impacted by COVID-19. Interruptions have led to heavy losses and the destruction of the entire produce of some farmers [13].

Additionally, during COVID-19, logistical impacts on international trade impacted the supply of food accessed internationally [14,15]. The reduced availability of imports

substantially increased the demand for locally produced food [16]. In response, farmers and those employed in associated sectors adopted various adaptations to mitigate the impacts of COVID-19 [17].

Recent systematic reviews related to the impacts of COVID-19 have included reviews focused on the impacts of COVID-19 on diet quality, food security, and nutrition in low- and middle-income countries [18]; the impact on the food supply chain [3]; supply chain management and development inspired by COVID-19 [19]; agricultural technological adaptation to the COVID-19 [20,21]; the immediate impacts of the first wave of the global pandemic on agricultural systems worldwide [22]; the transformation of the food sector, resilience, and security [23]; animal welfare and livestock supply chain [24]; frameworks of vulnerability, resilience, and risks [25]; food supply technology [21]; agrifood entrepreneurship [26]; food security in the first year of COVID-19 [27]; livestock systems and food security in low and middle-income countries [28]; impacts on African nations [29]; and global food security [30]. The systematic literature reviews cover many significant topics. However, none specifically focus on the smallholder farming sector. Publications by Asegie et al., 2021 [13] and Magar et al., 2020 [4] conclude that there needs more evidence of the systematic impacts of COVID-19 on the small-scale farming sector. Prior to any discussion on how smallholder farmers can improve their resilience to pandemics, there is a need for systematic evidence on the impacts, adaptations, barriers to adaptations, and recommendations concerning COVID-19 which policymakers, researchers, and NGOs may use to mitigate the impacts of future pandemics. This study will assist researchers in identifying gaps in the existing research on small-scale agriculture and "identify questions for which the existing evidence provides clear answers and thus for which further research is not necessary" [31]. We systematically document the available knowledge on COVID-19 impacts, adaptations, barriers to adaptations, policy suggestions, and research gaps. This study aims to:

(a)   Identify, document, and discuss COVID-19 impacts on the smallholder farming sector;
(b)   Identify, document, and discuss adaptation responses;
(c)   Identify, document, and discuss policy and research gaps regarding impacts and adaptations;
(d)   Identify, document, and discuss barriers to adaptations;
(e)   Commence a discussion on preparing for future pandemics based on the results section.

## 2. Research Methodology

We conducted a systematic literature review (SLR) using peer-reviewed literature published in four databases (Scopus, Web of Science, Agricola, PubMed) during 2020-2021. We followed the PRISMA guidelines for systematic review and used NVivo to analyse and code the studies.

### 2.1. Search Strategy

We tested initial search strings before determining the final string: "covid-19 OR corona OR covid19* AND smallholder* OR small-scale OR subsistence OR peasant." Literature that was either published or in the press was included. The search was limited to English publications.

### 2.2. Selection Phase

Publications were exported to 'EndNote' and were analysed to ensure the publications met all the criteria—this occurred after entering the query string into a relevant database. We defined the following exclusion criteria:

- Studies that lacked a clear focus on COVID-19.
- Studies that did not refer to smallholder farming (except in small-scale and large-scale cases).
- Studies about small-scale forestry and aquaculture (due to variations in production).

The initial search yielded 1790 studies—50 duplicated studies were removed, and 966 from irrelevant fields were excluded. The remaining 774 studies were screened. A rigorous screening process was applied—300 studies were read in full and were coded in NVIVO; 200 studies required us to read the abstract, introduction, discussion, and conclusions to decide if they should be included, and finally, 274 studies required us to read the title and abstract to decide if they should be included. Seven articles could not be retrieved. Fifty-three studies were used for the analysis. The screening technique was based on the 'Preferred Reporting Items for Systematic Reviews and Meta-Analyses' (PRISMA), 2020 (see Figure 1). (A table showing the 53 included articles can be found in Table S1 in the Supplementary Materials section).

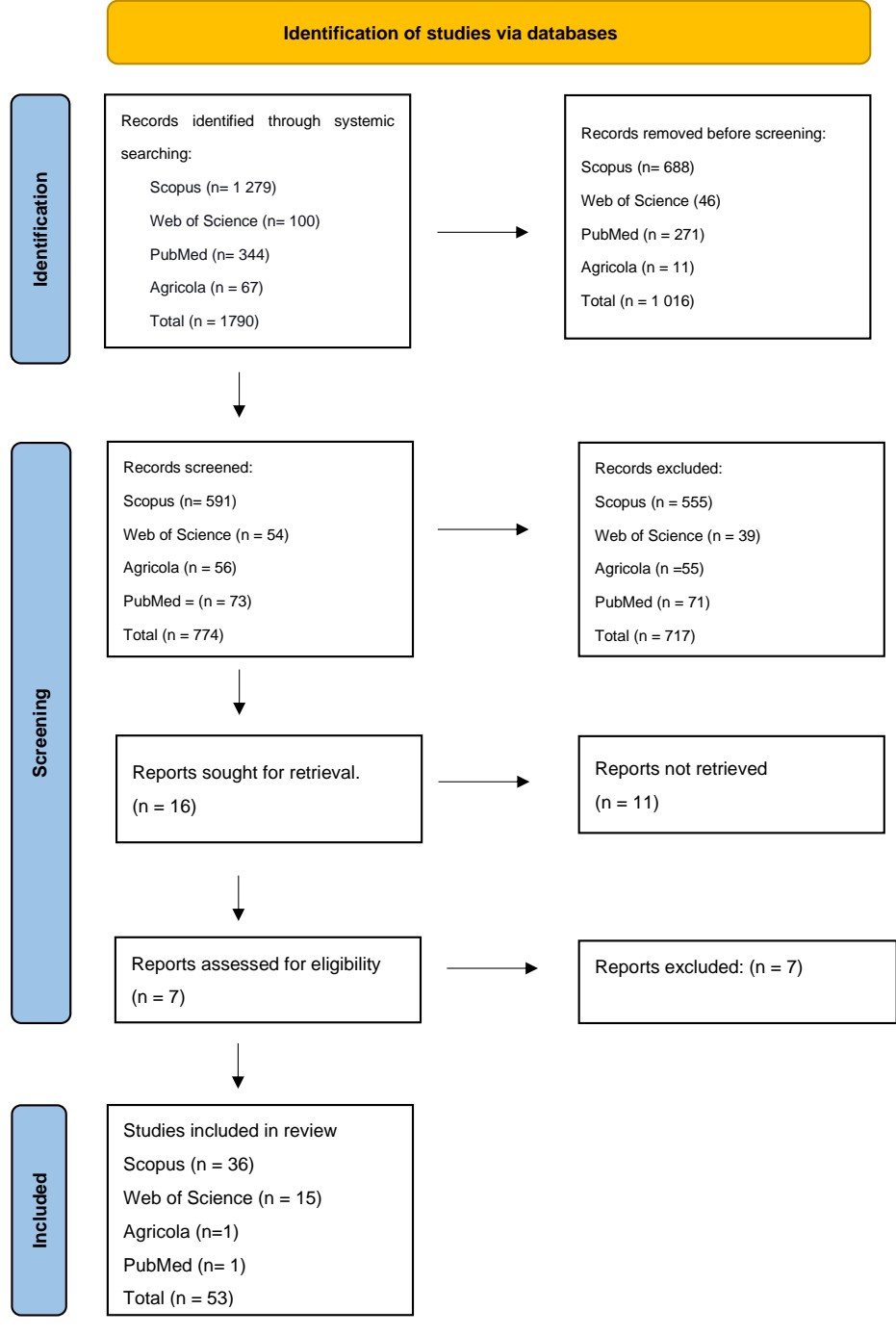

**Figure 1.** PRISMA 2020 flow diagram for SLR (included searches of databases only).

We included some studies that made a significant statement about smallholder farming during COVID-19, even if the main focus was not on smallholder farming.

*2.3. Analysis*

NVIVO was used to code the studies. NVIVO has been designed for researchers completing primary research or literature reviews. The program enables the coding of studies and notes, the completion of word searches, text mining, the construction of maps, and further exploration of nodes. See Figures 2 and 3. Extensive knowledge concerning impacts, adaptations, barriers to adaptations and policy, and research gaps were analysed and synthesised using NVIVO as our knowledge repository. Then, Excel was used to synthesise our analyses and the results further. The systematic review results section is based on NVIVO and Excel analyses. The discussion section is based on the results of the analyses documented in the results, insights, and suggestions based on our agricultural science experience.

Figure 2 provides a snapshot of the process. The themes and subthemes are shown as the features and drop menus available for coding and analysis.

Figure 3 shows the results of the text searches that we completed. The number of files that contained the search word or phrase was found in the files. The texts highlighted the word or phrase.

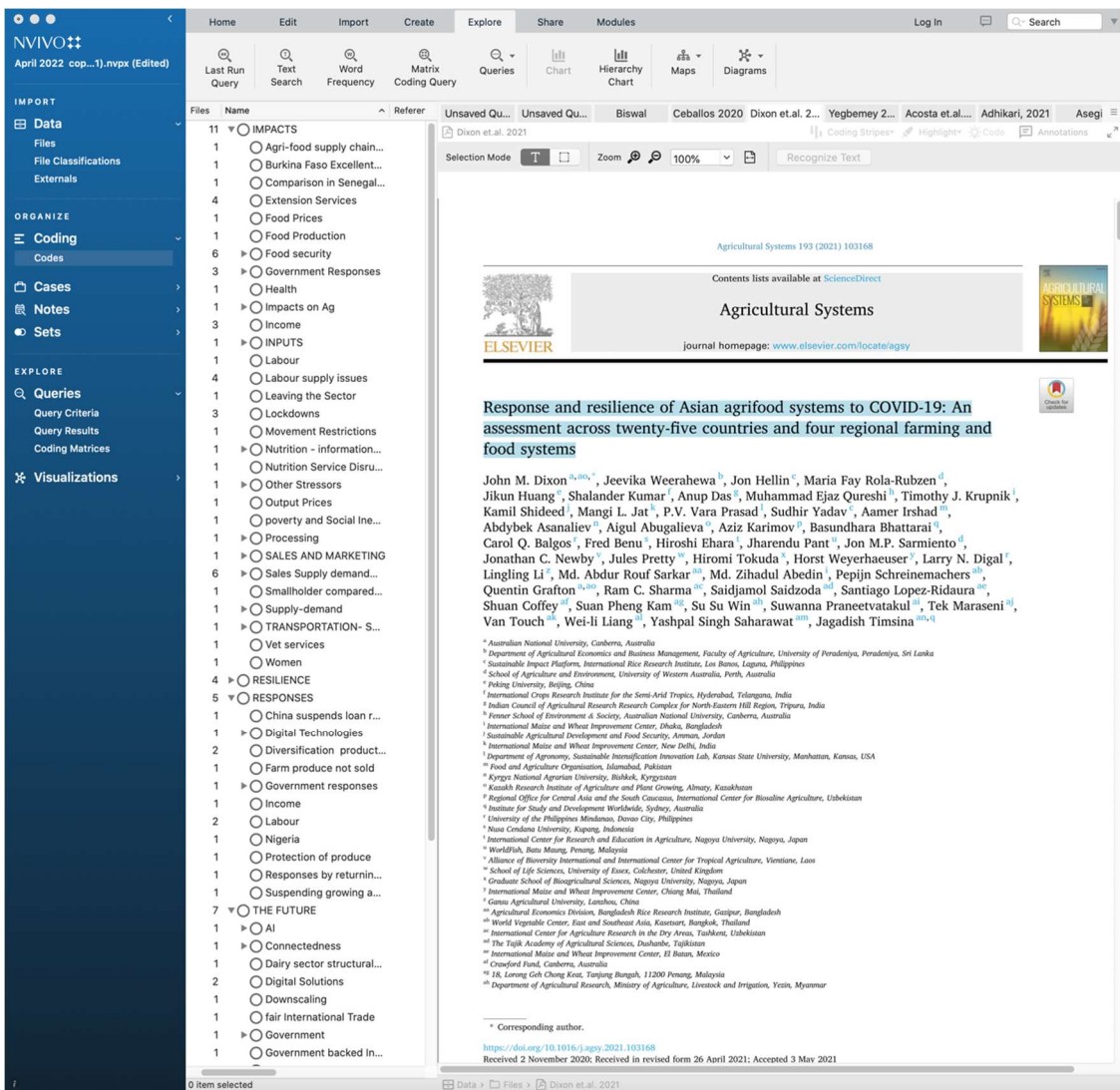

**Figure 2.** NVIVO code development.

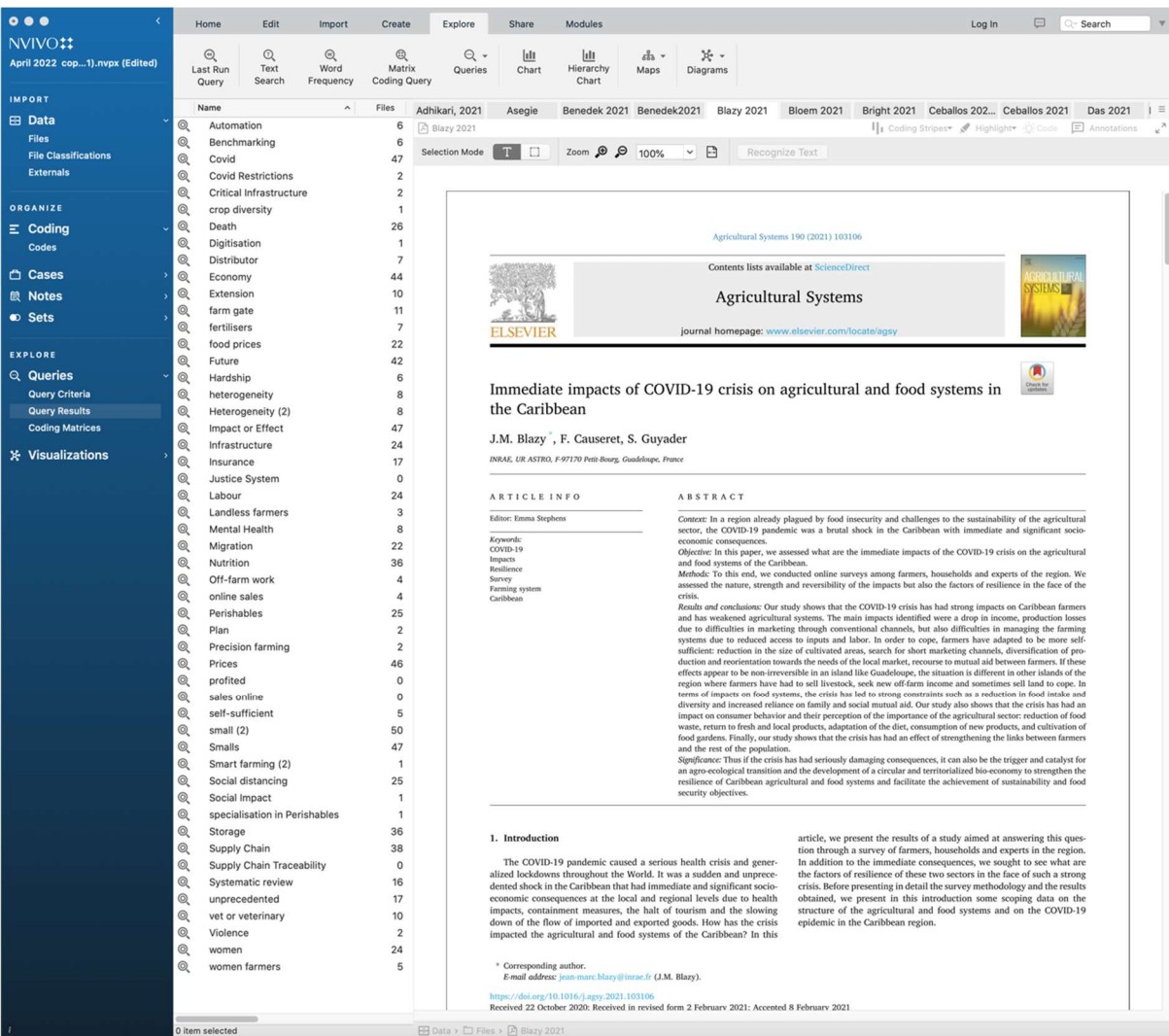

**Figure 3.** NVivo text searches.

## 3. Results

This Result (Section 3) documents the results using maps, tables, and texts. All the information in the results section was found while analysing publications in NVIVO and Excel. The discussion section (Section 4) discusses the results and insights of the authors of the systematic literature review. Sub-sections such as 4.5 (Options for Adaptations) discuss the results through the lens of what is required to ensure that smallholders will be resilient to future crises.

### 3.1. Description of Studies

In total, 25% of the publications were published in 2020 and 75% in 2021. The locations with the highest number of studies included India, Senegal, the Sub-Sahara, and Southeast Asian countries (see Figures 4 and 5).

We found a variety of data sources for secondary data, including reports, peer-reviewed studies, panels, the US Bureau of Statistics, blogs, media, and primary data sources, including farmers, NGO workers, government agencies, experts, academics, value chain agents, senior-level managers, agro-industrial companies, importers, and supply chain actors. In total, 68% of the data were from developing countries, 12% were from developed countries, and 20% were from a global perspective. In developing countries, there was more than twice the number of primary studies (227%) compared to secondary studies. See Table 1.

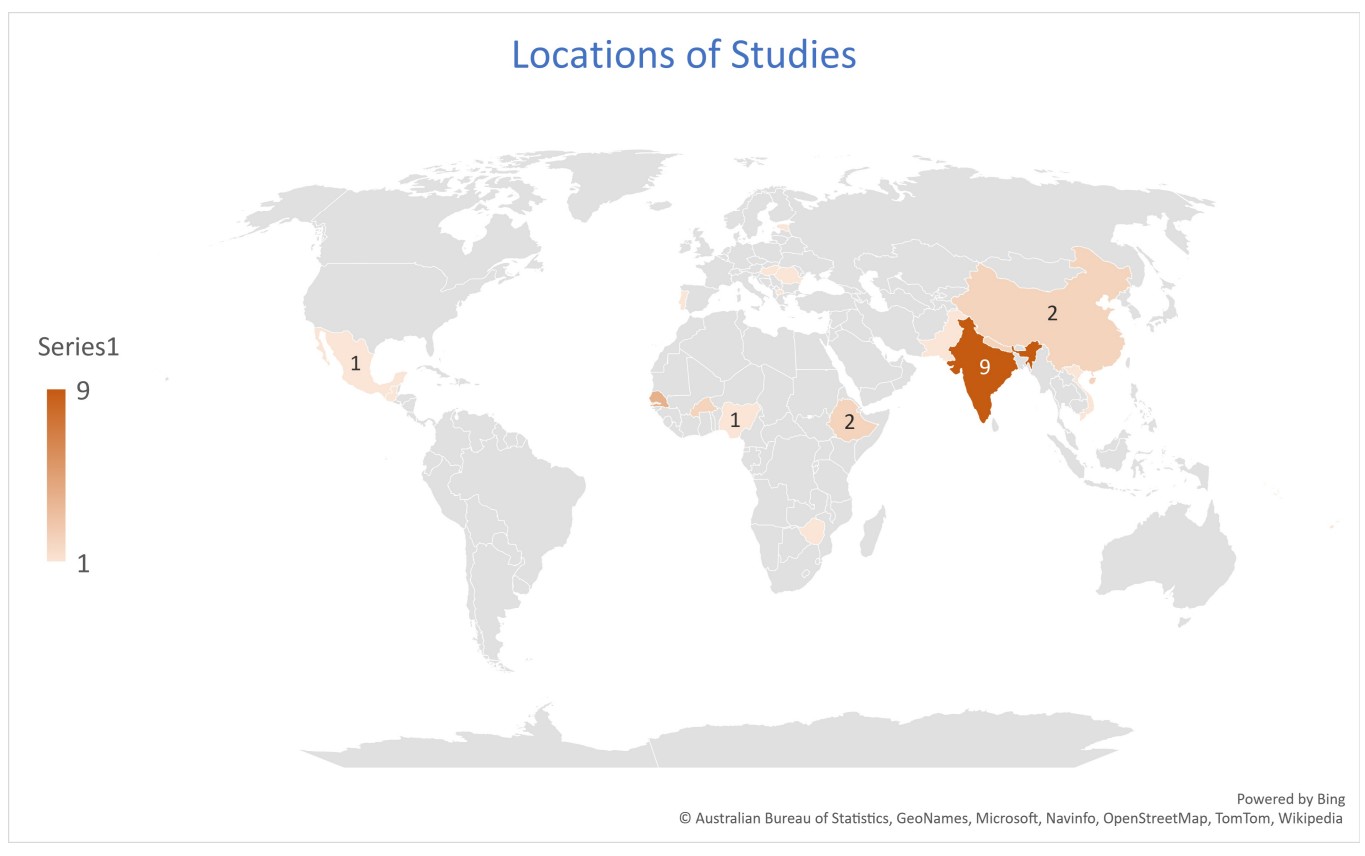

**Figure 4.** Map of locations of studies about smallholder farming (2020–2021). NB: Mouse-over enabled.

**Table 1.** Table displaying types of data at country and global level.

| Data Source | Primary Data in a Developing Country | Primary Data in Developed Country | Primary Data Global | Secondary Data Developing Country | Secondary Data Developed Country | Secondary Data Global |
|---|---|---|---|---|---|---|
| No. of publications | 25 | 3 | 7 | 11 | 3 | 4 |
| Percentage of publications | 47% | 5.60% | 7.50% | 20.80% | 5.60% | 13.21% |

There were seven primary studies with a global perspective (one comparative study and six subject reviews) and four secondary studies with a global perspective (review studies).

In a third of all the studies, the researchers used digital technologies to collect data, including phone and short text message surveys and online surveys. In contrast, in the other two-thirds of the studies, the researchers collected data using nondigital technology, including face-to-face surveys, interviews with farmers and key informants, panel discussions, and focus groups.

Most studies were about mixed farming (59%), while 26% were about crop farming and 15% were about livestock production. Thirty-one studies were about small-scale farming, and twenty-two were about a mix of small- and medium- or large-scale farming. The farm methods included organic farming (5 studies), traditional farming (1 study), climate-smart farming (3 studies), and conventional farming (1 study). However, 75% of the studies did not specify the farming method used.

In total, 57% were primary data, 34% were secondary data, and 9% were both.

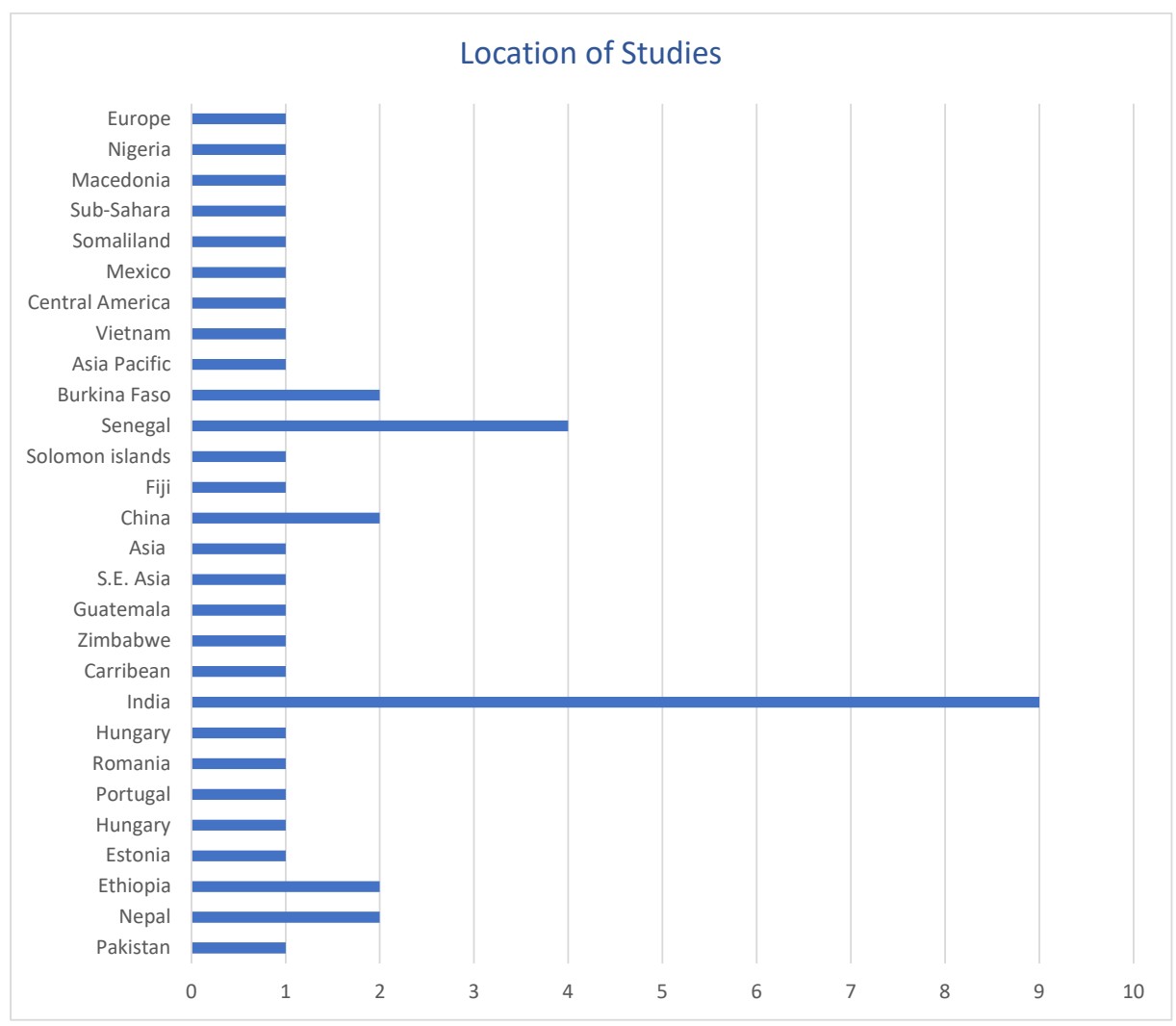

**Figure 5.** Locations of studies naming specific regions and counties.

### 3.2. Socioeconomic Impacts of COVID-19

Two main classifications in the research on the socioeconomic impacts of COVID-19 on small-scale farming were found: economic impacts on farm production and social impacts on farmers (see Table 2).

**Table 2.** Main impacts of COVID-19 on small-scale farmers and members of farm households.

|  | Main Impacts on Small-Scale Farming | Percentage of Studies Mentioning Impact |
|---|---|---|
| Economic (or farming) | Supply | 100% |
|  | Markets | 100% |
|  | Incomes | 96% |
|  | Labour | 94% |
|  | Access to farm Inputs | 81% |
|  | Production planning | 4% |
|  | Decline in Tourism | 25% |
| Social impacts on farmers or members of farm households. | Impacts on male farmers include increased workload. | 19% |
|  | Impacts on women include increased violence against women and reduced access to food. | 51% |
|  | Impacts on children include malnutrition and reduced access to education | 42% |
|  | Impacts on the aged include increased health vulnerability and reduced access to retail channels. | 23% |

The most mentioned impacts were disruption to supply (100% of studies) and disruptions to markets (100%), followed by labour shortages (94%) and interruptions to input supply (81%), and the least mentioned impact was production planning (4%).

### 3.2.1. Supply

All studies discussed impacts on supply, including supply chain interruption, interruptions to the supply of food to markets, processors and consumers, and the supply of farming inputs.

Agricultural supply chains were interrupted most severely during strict and prolonged lockdowns. In India, lockdowns caused enormous distress for poor and marginal farmers [8,12]. In China, where the government adopted a radical and aggressive approach to lockdowns in village areas, the impact on farmers was significantly severe in the short-term. The long-term impacts are unknown [32]. There were reports that all the food depots responsible for distributing food to people in Mugu district in Karnali, Nepal, had become empty of food stocks, and people could not buy food at all [6]: "There was a serious shortage of chemical fertilisers for rice because of transportation restrictions" in Nepal [6], and "The most significant effects on crop production were from a reduced labour supply, which significantly reduced access to agricultural inputs, lack of transport and sufficient markets for produce, due to the movement restrictions" [12].

### 3.2.2. Labour

In total, 50 of the 53 studies contained references to labour. The results indicate that the theme of labour was a significant factor. Several subthemes were found, including labour shortages [5,8,10]; accessing family labour [6,13]; severe impacts on labour-intensive farm production [10]; mobility restrictions [5]; morbidity [5]; labour migration, including reverse migration [6]; diversification to accommodate labour shortages [16]; the severe impact on landless labourers [11,33]; and labour-saving technologies [6].

### 3.2.3. Inputs

Twenty-three studies contained references concerning production inputs. These results indicate that the supply of "inputs" significantly impacted some small-scale farms. Significant references to several subthemes were found. The themes included the subsidisation of inputs [34]; the access and delivery of inputs [6,35]; new technologies to enable the payment of inputs [10]; seed shortages [10,36,37]; a shortage of fertilisers [14]; reluctance to invest in inputs [11]; inputs for crop protection [38]; input price increases [39]; locally produced input networks [7]; the quality of inputs [6]; and fodder inputs [8]. The results indicate severe interruption to accessing inputs for many small-scale farmers. Finally the direct of supply of inputs may be required [40].

### 3.2.4. Incomes

Many smallholder farmer incomes were severely impacted by COVID-19. Ninety-six percent of the publications reviewed discussed the impact on income. The economic impact that COVID-19 had on income further impacted the social impacts by reducing income to purchase food and other necessities.

### 3.2.5. Markets

All studies in the SLR referred to markets, indicating severe impact on markets during COVID-19. During the COVID-19, lockdowns significantly impeded access to markets. Many markets closed [16,40,41], and others experienced shortages of available products [10]. Markets and farmers experienced price changes due to the slowing down of supply [10].

The tourism market was severely interrupted due to international travel bans. The interruption impacted farmers selling to the tourist market [41]. In Central America and Mexico, grains, vegetables, fruits, roots, tubers, and meat producers were most affected due to tourism [41].

### 3.2.6. Production Planning and Timing of Farm Activities

According to Abid and Jie [10], lockdowns that coincided with harvesting led to the destruction of crops. Lockdowns that coincided with seeding meant farmers had to postpone seeding activities and face a loss of income. Further, according to Ceballos et al. [42], COVID-19 severely disrupted farm production plans, and farmers needed help executing essential farming activities to schedule. The affected farmers may not have been able to harvest, fertilise crops, plant seeds, or control pests, depending on the stage in which the lockdowns occurred. In India, the distribution of farm produce during lockdown periods became impossible due to travel restrictions [42]. In Odisha and Haryana, India, farmers adjusted the timing of harvests. However, traders were unable to purchase them at the farm gate. The absence of farmgate sales significantly impacted the sale of wheat. Farmers could only harvest and distribute wheat after the lockdown commenced [42].

### 3.2.7. The Decline in Tourism

Twenty-five percent of publications discuss how smallholder farmers were impacted by the decline in tourism due to COVID-19. The tourism sector closed dramatically in many popular destinations. Smallholder farmers supplying markets were severely impacted.

### 3.3. Social Aspects

The literature contained significant research on the impacts on women, children, and the elderly. References to males were found in 10 studies. However, when males were mentioned, it was usually used in comparison to females. Middendorf et al. [43] were the only researchers to examine the social impacts of COVID-19 on males.

### 3.3.1. Women

Violence against women increased during COVID-19 [6]. The division of household food has been unequal during COVID-19, with women receiving a lesser share of the available food [6]. Malnutrition increased for women due to COVID-19 lockdowns [10], and women's household duties increased due to school closures. Many women already responsible for household duties performed extra work, including farm work, selling produce at markets, and educating the children [6,11].

### 3.3.2. Children

COVID-19 exasperated food and nutritional problems experienced by children in farm households [14]. Deep concerns exist about the increasing malnutrition, wasting, and stunting of children in developing countries [41,44]. Child abuse increased during COVID-19. Kang et al. [2] called for an increase in child protection and education services for children.

### 3.3.3. The Aged

The elderly were called upon to perform farm work during the COVID-19 due to labour shortages [6]. The elderly found it difficult to access mainstream retail channels [38]. The elderly were more vulnerable to COVID-19, which may have made it necessary for other family members to perform some of the elderly persons' duties [45].

### 3.3.4. Men

Male farmers have consistently reported increased workloads since the start of COVID-19, with many working alone [43]. Middendorf's study [43] on vegetable production in Burkina Faso found that workloads for men increased by an average of 16.8% and that 31% were unable to hire the labour required for farming.

### 3.3.5. Minority Groups

There appears to be a gap in the research concerning social impacts on minority groups living on smallholder farms, including people with disabilities, minority religious groups, and those with alternative sexualities.

### 3.3.6. Incomes

The social impacts of reduced or insufficient income were not found in the SLR publications. However, relevant factors such as reduced incomes and increased working hours shown in the results would have had a negative effect on relationships and mental health.

### 3.4. Adaptations

Forty (75.47%) studies referred to adaptations, indicating that adaptations have been a significant focus of the research. The main adaptation strategies are to access family labour, negotiate labour with locals, use digital phone technology, use other digital communication strategies, diversify products, and diversify markets (see Table 3).

**Table 3.** Adaptations mentioned and the percentages of articles in which they are found.

| Adaptation Strategy | Studies Mentioning Strategy (%) |
| --- | --- |
| Technological | 83% N = 44 |
| Communication strategies | 62% N = 33 |
| Increasing Storage facilities | 72% N = 38 |
| Diversifying produce, markets, or income sources | 34% N = 18 |
| Increasing access to family labour | 18% N = 9 |
| Use of mobile phones or cellular phones | 8% N = 4 |
| Negotiating labour in the local community | 4% N = 2 |

### 3.4.1. Technology

There has been a significant interest in technological innovations to assist farmers in coping with supply issues [46]. The technologies under research include artificial intelligence, big data, blockchain, cloud computing, the internet of things, and machine learning. Developing and emerging technologies possess enormous potential to improve production yields and soils, synthesise farm data, efficiently use fertilisers, and mitigate the impacts of disasters [46].

### 3.4.2. Communication

The theme of communication is discussed in 33 of the 53 studies, indicating that this is a significant adaptation. Many farmers utilise digital communication technologies to maintain the feasibility of their small-scale farms [41]. Communication technologies enabled farmers to make online purchases and sales when face-to-face communication was a high risk [47]. Digital platforms enable communication to strengthen buyer–supplier collaboration and lower food security and food shortage risks [10].

### 3.4.3. Digital phones

The use of digital phones and digitised phone platforms has increased significantly since the beginning of COVID-19 [44,48]. Mobile phones have enabled access to agricultural technologies and information regarding farming methods [44,48] Technologies accessible on phones can assist farmers with water optimisation, the appropriation of seeds, and determining optimum fertiliser levels and pesticide use [48]. As mobile phone infrastructure expands and phone ownership increases, mobile phone technology will significantly increase yields and reduce costs. The use of digital phones will dramatically enhance communication; however, in some remote areas, digital phone infrastructure is still required.

### 3.4.4. Diversification of Produce

Farmers who diversified production were more effective in adapting to the COVID-19 crisis. Farmers started small-scale diversification with vegetables [36]. To cope, farmers diversified production to meet the local market's needs [49].

### 3.4.5. Diversifying Markets

Farmers sought short marketing channels [49]. Some farmers diversified distribution channels and removed supply chain intermediaries [47].

### 3.4.6. Diversifying Income Streams

The diversification of income streams allowed farmers to act more flexibly [41]. Farmers could either increase or decrease farm activity. However, reliance on income streams other than farming may have had a negative effect if these streams were interrupted [10].

### 3.4.7. Storage

Due to the impacts on logistics, storage facilities were desperately required, especially for perishables that required cold storage facilities [1,50].

### 3.4.8. Accessing Family and Local Community Labour

In response to labour shortages, many farmers' adapted by accessing family labour and exchange labour with the local community [36].

### 3.5. Barriers to Adaptations

There were eight barriers to implementation documented in the review. These included the lack of government support to develop new technologies [10], a lack of information [8], the price of phone data [49], infrastructure [12], access to the internet [48], research that compares perceived impacts and actual impacts [48], AI (artificial intelligence) is in its early stages [46], and the inelasticity of demand for produce, meant farmers were unable to increase sales by reducing prices [51].

### 3.6. Policy

The publications either discussed the impact of policy or were concerned with developing policies to mitigate the impacts of COVID-19. Initially, governments developed policies to limit the spread of COVID-19 [1,2]. Policy restrictions severely impacted many farmers, and food shortages were imminent. Governments relaxed policy restrictions for critical economic sectors to ensure the supply of food and other essential services.

COVID-19 was an unprecedented event, and many smallholder farmers could not cope. Many researchers in this review developed policy recommendations to address the impacts on farmers. Eighty-seven percent of the studies included policy recommendations. Table 4 shows the target audience for different categories of recommendations. The percentages are calculated by assessing which audience is most likely responsible for developing or implementing the policy. Table 4, therefore, contains general estimates.

**Table 4.** Target audiences of recommendations.

| Target Audience | Recommendations (% est.) |
|---|---|
| Government | 31% |
| Policy Makers | 21% |
| Research Community | 6% |
| NGO | 2% |
| Farmers | 11% |
| Agribusiness | 2% |
| Technologists | 2% |
| Insufficient evidence | 23% |

We aim to document the results concerning policymaking in the results concisely. The discussion section discusses the results of developing a policy for future pandemic crises.

### 3.6.1. Demands in Governmental Policy Change

Policy recommendations directly addressed to governments by the authors are documented. Nchanji and Lutomia [39], in a broad statement, recommend that governments develop a range of policy interventions to revitalise and improve farmers' resilience [39]. Other authors provide the finer details about the actual policy. Family support, through various strategies, is a priority of many researchers. Hirvonen [51] espouses that governments need to ensure the availability of nutritious foods. Asegie et al. [13] recommend that governments focus on immediate and long-term intervention strategies to assist the most impacted families using social security and revolve funding mechanisms. Kang et al. [2] recommend that governments provide social security and a social security net. Nchanji et al. [39] call for livelihood relief. Abid and Jie [10] propose that governments invest in new technologies, sustain the flow of agricultural products along the supply chain, encourage banks to create easy and quick transaction methods, control food security and price, and launch commercial transaction apps. Adhikari [6] calls for governments to increase the mechanisation of farms and develop digitisation. Moreover, they call for an increase in food reserves and state that governments should provide smallholder farmers with cash support to develop fallow land. Biswal et al. [8] recommend that governments research all forms of farming to acquire the holistic information required to attract support for all sectors and revive value chains. Dilnashin et al. [12] call for governments to purchase a surplus product and to avoid export bans and import restrictions. They also call for mobility restrictions imposed by governments to reduce the spread of COVID-19 to be lifted to allow farmers access to markets. DU et al. [32] recommend that governments provide vocational education and training for family farm owners; develop a policy and market environment supporting the long-term, stable operation of family farms; and improve the agricultural insurance market by making it accessible to more agents. Haggag [48] recommends that governments intervene in predatory price increases. Ijaz et al. [50] recommend increased communication to ensure the continuous flow of inputs and outputs. Nchanji et al. [40] call for governments to support actors across the food supply system with input subsidies, livelihood relief, supporting innovation, further digitisation, and supporting bulk purchasing. Magar et al. [4] recommend that government assists farmers with technical services by institutionalising research and extension services and improving government coordination within the tiers of government.

### 3.6.2. Policy Directed at Policymakers

There are two direct policy recommendations for policymakers discussed in the review. However, many policy recommendations will become the policymakers' responsibility in practice. Meuwissen et al. [52] call for policymakers to decide whether regional and short-value chains are more resilient than transnational ones. They also call for policies to strengthen anticipatory capacities, consider whether connectedness in value chains and diversity can be integral to policy design, develop stress tests, and provide the resources necessary to enable the transition to maintain public goods and services. Yegbemey et al. [53] call for policymakers to design strategies to encourage farmers to adopt innovative market-oriented strategies with the assistance of government extension services.

### 3.6.3. Policy Directed at NGOs

There is one publication that recommends NGO policy development. Asegie et al. [13] recommend that government donor organisations provide immediate and long-term intervention strategies through the implementation of social security programs and revolve funding.

### 3.6.4. Policy Directed at farmers

Huss recommends that farmers build or invest in onsite storage facilities [1]. Ijaz [50] recommends that livestock farmers should first communicate with suppliers of consumables, feed distributors, and professionals such as veterinarians and meat processors to find solutions to secure inputs supply, farm services, and the meat supply chain. Secondly, farmers should talk with farmers associations to reach out to the policymakers to obtain compulsory exemptions for the transportation of feed, animals, and personnel. Thirdly, farmers should adopt strict precautionary and management measures at farms to avoid disease spread [2]. Benedek et al. [16] propose that farmers increase produce diversity.

### 3.6.5. Policy Directed at the Research Community

Thulasiraman [54] calls for focused research towards local food chain sectors, the development and revival of traditional food sectors, and technological interventions of traditional knowledge. Informal food sectors such as raw milk trading, jaggery production, and the cultivation and preservation of indigenous fruits, vegetables, and crop varieties offer excellent employment opportunities and considerable food availability when operated on a small scale. Nchanji and Lutomia [39] call for Agricultural Research Systems to support SMEs in developing innovative business models that enable them to cope with the effects of any ensuing pandemic.

### 3.6.6. Policy Directed at Agribusiness

Goswami suggests that agricultural cooperatives, SHG, or existing federal programs extend credit to smallholder farmers. Magar et al. [4] suggest that agribusiness adapts and supports farmers during a pandemic.

### 3.6.7. Policy Directed at Technologists

Thulasiraman [54] calls for focused research towards technological interventions to be used in pandemic crises. (A table summariising policies and adaptations can be found in Table S2 in the Supplementary Materials section).

## 4. Discussion

### 4.1. Insights from Previous Pandemics

Before discussing the significance of the results for future pandemics and what activities we would support concerning future pandemic crises, we discuss the impacts of pandemics pre-COVID and discuss and compare current smallholder farmers' adaptations with past adaptations.

Previous pandemics, including MERS-COV, SARS 2002-2004, the Avian Bird Flu, and the Ebola Virus, did not have the global impact of COVID-19. However, the impacts on smallholder farming were significant [55]. According to Gatiso et al., the impact of Ebola on agriculture and livelihoods was unequivocal evidence that epidemics adversely impact livelihoods [55]. Gatiso et al. found that the impacts may have long-lasting effects on livelihoods, even for those not directly impacted [55]. Feuerbacher et al. reported that infectious disease depletes farmers' access to their primary asset, labour, due to premature deaths and morbidity [56]. Muteia et al. found that 21% of Nigerian smallholder poultry farmers lost between 80-100% of their income [57]. Considering the evidence, we assume that future pandemics will affect smallholder farmers.

The literature on adaptations for smallholder farmers before COVID-19 was comprehensive; however, previous adaptations are less relevant to the impacts of COVID-19. Our analysis of the impacts showed that the main differences during COVID-19 include personal and workers' health and safety, lockdowns, supply, tourism, and international trade.

There has been recent significant interest in the literature concerning smallholder farm adaptations for climate change. However, research concerning adaptations required

for pandemics should have also been a priority due to previous evidence concerning the adverse effects of past pandemics on smallholder farmers [56].

Previously, smallholder farmers in developing counties have used traditional adaptations such as accessing assistance from their local community [55]. They could not access modern adaptations due to personal financial constraints and the lack of access to credit [55]. However, the analysis of the current adaptations showed that some farmers in developing countries are applying modern technologies. For example, our analysis of online communication strategies showed that many smallholder farmers are using digital telephones. As a result, many smallholder farmers in developing countries can obtain farm management information using their mobile phones. Additionally, many farmers now belong to online communities and can discuss farm management issues with other farmers.

COVID-19 has led to a significant increase in research about the impacts on smallholder farmers and their adaptations during pandemics. However, we found that measurements of both the impacts and adaptations were previously, and continue to be, severely limited. Birthal et al. [58] recognised the importance of empirical evidence and descriptive and inferential statistical methods applicable to impacts and adaptations. They applied these to their study on smallholder farming.

Finally, we assume that preparation for all future pandemics is impossible due to the uniqueness of pandemics. However, we believe that the information from this systematic review is invaluable to future policies to mitigate the impacts of pandemics on smallholder farmers. Further, pre-COVID research on adaptations is also invaluable.

### 4.2. Description of Studies

#### 4.2.1. Years of Publication

The small number of publications in 2020 could be due to travel restrictions and researchers' concerns about completing the research. Our view is that future unprecedented pandemics may result in a similar experience. Furthermore, future pandemics that are more dangerous than COVID-19 could seriously prohibit research activity, particularly during the early stages.

#### 4.2.2. Locations of Studies

Research during COVID-19 was mainly conducted in the Sub-Sahara region of Africa, SE Asia, and China. We espouse that researchers should have specified why the research was undertaken in a specific region. We can speculate that researchers are interested in researching developing regions as they are home to a large cohort of small-scale farmers. There was broad concern from humanitarian organisations that smallholder family households may return to poverty in these regions. There may be other reasons why little research was completed in other regions. This may be because the smallholder agricultural sector was marginal in some regions or the spread of COVID-19 was less severe or even non-existent.

Furthermore, many countries have social welfare nets that protect smallholder farmers, thus making the need for research less urgent. The main problem concerning mitigating future pandemics is that impacts may be more severe than those of COVID-19, which could threaten food security. The adaptations required will depend on the nature of the pandemic. Regions previously unaffected may still be impacted in the future. Furthermore, previously unaffected regions may need to take adequate precautions. We recommend that any country with a significant smallholder farming sector be prepared for all forms of future pandemics.

#### 4.2.3. Data

The data analysis showed that primary data far outweighed secondary data. Considering the dangers of conducting research, this result was unexpected. A significant result was that two-thirds of the data were obtained from face-to-face surveys, interviews with farmers and key informants, panel discussions, and focus groups. Concerning future

pandemics that prove more lethal and contagious than COVID-19, it may not be possible to gain any of these types of data, which would severely affect attempts to access vital data. Researchers must use safe methods to provide good data before conducting research. Farmers in areas where it is impossible to access digital data or farmers who cannot afford digital data may be severely affected. The result may leave smallholder farmers in the dark when attempting to access vital knowledge related to their family household and farming needs. Communicating research findings and participating in research that benefits smallholder farmers are additional reasons why all smallholder farmers should have access to mobile phones.

### *4.3. Economic Impacts*

Economic impacts, including those on supply, markets, incomes, labour availability, and access to farm inputs, as well as interruptions to production planning and the decline in tourism, are all impacts that may have been severe for most or many smallholder farmers. The economic impacts found in the results need to be included in policies for future pandemics. The adaptations suggested for each are discussed in the following section.

#### 4.3.1. Supply

Supply systems were severely interrupted initially [8,12] and during prolonged lockdowns, and this forced farmers to adapt and change production plans [10], which included the diversification of produce [36,49], diversifying markets, or suspending production [42]. Supply systems include farming input supplies and supplies to markets and processors. Some smallholder farmers lost their entire livelihood due to supply. Supply may be the most crucial factor in future pandemics, but fortunately, new technologies are under development, and logistic practices are continuously improving. Future supply logistics are likely to be far more sophisticated than the supply system of the COVID-19 era.

#### 4.3.2. Markets

The results showed that lockdowns significantly impeded market access [10,16,40,41,59] including closing or having supply shortages. Markets also experienced price changes. A policy that ensures fair prices during a pandemic may help in future pandemics. Additionally, the government should address predatory pricing policies [48].

#### 4.3.3. Incomes

Despite several media publications stating the agricultural sector was the least impacted economic sector, only decreasing production by three percent, these data were not specific to smallholder farming. The impact on the smallholder sector was likely to be far greater than three percent. To prepare for future pandemics, accurate results are required. More accurate information could be found if governments conducted a special census on COVID-19. While it is unlikely that they will conduct a special census, taxation records might assist in assessing the income impacts of COVID-19 on smallholder farmers.

#### 4.3.4. Labour

The results showed that farmers found difficulties obtaining labour [5,8,10]. Adaptations to access labour were limited to accessing family and community members wishing to take the risk, probably because they needed to feed their families. We recommend governments develop employment services and safe transport options for the transmigratory movement of immigrant workers. Farmers can prepare for future pandemics by building safe buildings for their workers.

#### 4.3.5. Access to Farm Inputs

Access to farming inputs is a critical aspect of the supply system for smallholder farmers. Shortages of fertiliser [14] and seed [10,36,37] and fodder [8], and farmers' ability to store quality inputs must be addressed in policy for future pandemics. We recommend

adaptations that use farm resources, such as producing fertilisers or producing seeds. Supplying seeds for personal use can be easily accomplished with crop seeds such as corn, rice, and wheat.

### 4.3.6. Production Planning

Two publications from the review discussed production planning [10,42]. This was likely due to more excellent farming knowledge or a more rigorous approach by the researchers. Production planning must be included in future pandemic policies so farmers can be better prepared and respond by making changes more flexibly.

### 4.3.7. Impacts on Tourism

Many smallholder farmers supply to local markets; however, a significant number supply to the tourist market [41]. Future pandemics, at least as contagious and lethal as COVID-19, will likely result in closures to tourist destinations. Smallholder farmers may need to rely on subsistence production and look for other sources of income. Others may survive or help others survive by sharing produce within their local community. We support and recommend developing community-based support groups to help others during a crisis.

### 4.4. Social Impacts

Policies to mitigate the social impacts are also required, as outlined in the discussion below.

### 4.4.1. Impacts on Women

The impacts showed an increased level of violence toward women [6], an increase in malnutrition [10], and an increase in workloads [6,11]. Another factor that may have impacted women is a decreased income, causing marital disputes and increased pressure on women (and others). We support the development of services for women experiencing violence or mental health issues and support these services being boosted during times of pandemics. We also support government efforts to increase awareness and the use of social media as new pandemics arise.

### 4.4.2. Impacts on Children

During COVID-19, there was an increase in child abuse [2] and malnutrition [14] among children. We strongly support the development of government services to monitor and act on child abuse and the development services to monitor children's diets and ensure they receive nutritious and adequate meals.

### 4.4.3. Impacts on the Aged

The aged were more vulnerable to COVID-19. Some were less able to assist with household work [45], but other aged persons were called upon to assist by helping with household work [6]. Some of the aged were less able to access food from markets and access services [38]. We support the development of increased aged services to support the elderly and the preparation of services and policies for the aged during pandemics.

### 4.4.4. Impacts on Males

Generally, males were better able to cope with COVID-19. Many men worked more hours during COVID-19 [43]. We support services for men unable to cope during pandemics, even though men may be reluctant to seek these services. Men's use of services should be monitored and resourced as needed.

### 4.4.5. Impacts on Minority Groups

The results showed a research gap concerning minority groups. Minority groups may have been excluded because they cannot communicate their needs. We support the

development of research to identify minority groups and their experience during COVID-19. If it is found that minority groups cannot communicate their needs, we support the development of services that enable minority groups to do so.

### 4.4.6. The Social Impact of Reduced Income

The domestic tensions caused by a lack of income may increase domestic violence. We recommend that all governments provide safety nets for citizens who fail to provide adequately for their families and themselves. Furthermore, a person's capacity to work is often misjudged. People with disabilities may look able to work but cannot work full-time. We recommend government policies to assist all people with inadequate incomes to achieve a good quality of life, thus reducing household stress and domestic violence.

### 4.5. Options for Adaptations

There were eight significant adaptations documented within the results section of the review. Policymakers should identify the adaptations that can be easily implemented in the region they are developing and consider strategies to enable adaptations with significant barriers.

### 4.5.1. Technology

New technologies can transform smallholder agricultural systems. This may result in higher profits for farmers and smallholder farmers, enabling efficient supply systems during pandemics and the automation of production processes. Many farmers are already taking advantage of new technologies. Automatic computer-controlled farms already exist. This offers significant opportunities for smallholder farmers. Smallholder farmers could combine farms by sharing the costs and benefits of automated farming. Access to additional finances is improving, and financial institutions will be able to see the benefit for farmers and their companies. Some smallholder farmers may view this as a pipe dream. However, if a suitable model is produced locally or even nationally, they will see the benefits and may come on board. Automatic farms can significantly reduce the effects of pandemics on smallholder farmers. There would be less or no need for labour, and produce would be safe, clean, and disease-free with onsite testing. We recommend developing technology to create noncontact agricultural systems, including using automated vehicles to supply produce during pandemic events.

### 4.5.2. Communications

The results showed that developing communication technologies could assist many smallholder farmers [10,41,47]. Half a century ago, many of these communication technologies were non-existent. Older farmers who have watched and adapted to these new communications and younger farmers who have learnt these skills at school now have the skills to conduct most of their marketing and sales, observe daily changes in daily prices, order inputs, and obtain essential information relevant to their business needs. Younger farmers who have learnt these skills could find supplementary work by assisting older farmers lacking the newer communication skills. The communication strategies will assist farmers in confidently growing produce for local demand, locating processors, negotiating prices, organising safe and personal delivery, and creating online shops. Farmers with the necessary skills will be well positioned to negotiate better deals with intermediaries because of their power to negotiate online directly with processors, access a wide range of markets, and find new local customers. With new computers, such as Chromebooks, available at a fraction of the cost of regular computers, more smallholder farmers can afford to take advantage of new communications. Ideally, smallholder farmers need both a low-cost computer and a digital phone to take full advantage of the new opportunities available to them. We recommend that further education be provided for all farmers who still need to develop skills in communicating online.

### 4.5.3. Digital Phones

Digital phones were beneficial to smallholder farmers during COVID-19. Minicomputer digital phones can be carried in your pocket, making it possible to access information, receive calls, and conduct business at all hours. Smallholder farmers can use digital phones to access the information sites necessary to run their businesses and keep up with local news concerning lockdowns and transport restrictions [44,48]. The use of digital phones has been proven to be very useful during COVID-19, and farmers should make every effort to purchase a digital phone if the necessary infrastructure is available. Further, it recommended that all governments provide digital phone infrastructure. Digital phones are of even greater value during a pandemic crisis when face-to-face services are unavailable due to restrictions.

### 4.5.4. Diversification of Produce

The results relating to the diversification of products have proven to be effective [36]; however, the ability to make the right decisions regarding which crops to diversify is complex, and farmers will require assistance from agricultural extension services to ensure the soil and climate is suitable. Smallholder farmers will require assistance to learn how and when to grow crops they have yet to grow. However, with the right choices and education, diversification is relatively implementable by smallholder farmers. To prepare for future pandemics, we suggest extension services encourage and assist farmers in growing new crops and produce and keep continuous records to ensure the right choices are made during a pandemic. An obvious choice for farmers is growing non-perishable crops during periods when supply is severely interrupted.

### 4.5.5. Diversifying Markets

Diversifying markets creates more opportunities for smallholder farmers to sell their produce. If dealing with an intermediary who becomes unable to work due to illness or dies, the smallholder farmer has no choice but to explore new markets or lose most or all their produce. Smallholder farmers with market options will gain leverage in the market and will be able to negotiate fairer prices. Additionally, shorter distances to travel will reduce the risk of being exposed to a pandemic.

### 4.5.6. Diversifying Income Streams

Diversifying income streams enables smallholder farmers to supply family needs and finance their farms [41]. The ability to finance farm operations after a farming crisis would be invaluable to smallholder families. However, many smallholder farms lost additional income streams during COVID-19. There has been considerable interest in enabling smallholder farmers to access credit, and this looks set to improve.

### 4.5.7. Storage Facilities

Storage is critical for smallholder farmers. Cold storage is desperately required for growers of perishable produce such as fruits and vegetables [1,50]. The government has already commenced a storage program for smallholder farmers in Indonesia. All governments without adequate storage for the smallholder agricultural sector should follow Indonesia's lead.

### 4.5.8. Accessing Family and Community Labour

Accessing family and local labour is common for smallholder farmers in many countries [36], particularly during seeding and harvest. Both seeding and harvesting can be completed in 2 to 3 days on small farms. Smallholder farmers accessing local labour will not need to house their workers.

*4.6. Barriers to Adaptations*

Eight adaptations were found in the review, all of which need to be addressed by the government [8,10,12,46,48,49,51]. A government's ability to address these barriers may be limited or impossible due to financial and physical constraints. Many developing countries cannot and have never, in the past, been able to afford to assist with these adaptations, such as developing new technologies and providing roads, storage, and digital phone infrastructure. For example, in Indonesia, where approximately 50% of Indonesian farmers are smallholders, there are over 6000 inhabited islands, many of which are difficult or impossible to service.

Additionally, many developing and developed countries have suffered huge debts due to the impacts of COVID-19, limiting the support governments have been able to give to smallholder farmers and efforts to become more resilient to pandemics. The solution for farmers unable to benefit from the adaptations with significant barriers will be to either operate their farms as usual, if possible, or use those adaptations without barriers, such as the diversification of products, making sales locally, or looking for additional work.

*4.7. Policy for the Future*

The results have shown which policymaking group should be responsible for each policy aspect. The following discussion aims to provide practical policymaking advice based on our results and knowledge of agricultural science. The aim is to discuss policies for future pandemics.

### 4.7.1. Government Policymaking

The results section of this review showed that researchers have been able to specify significant policy recommendations for governments. The recommendations for the government will be of interest to government-employed policymakers. The results indicate that family support, including cash payments and providing nutritious food for families, is a significant priority of those researchers making policy recommendations for the government [2,39,52]. Farming support through cash and providing seed and fertiliser to keep the farms operating should also be a priority. Government research, technological development, and the provision of storage facilities should be priorities for governments because of the substantial impacts on supply that severely impact smallholder families.

### 4.7.2. Policy Development for Policymakers

Policymakers exist within governments, research institutions, and business organisations; policymakers should include farmers and others working in the smallholder agricultural sector. As stakeholders', farmers, transport workers, suppliers, and processors will have expert knowledge not available through other sources. Farmers have specialised experience in growing and knowledge regarding when to plant and what to plant. Policymakers who work with and engage with the smallholder agricultural sector will have access to insider knowledge that can improve policies for future pandemics. Agribusiness and farmers included in policymaking will be more likely to adopt policy suggestions directed at them.

### 4.7.3. NGO Policymaking

NGOs' strategy to support smallholders is understudied. Only one author discusses NGOs [13]. We support NGOs' engagement in family and farm management support. Additionally, research by NGOs will further enable NGOs to assist smallholder farmers in achieving resilience during pandemic crises.

### 4.7.4. Policy for Smallholder Farmers

Three policies documented in the results were directed at farmers concerning building storage, improving their communications, and diversifying produce [1,4,50]. Smallholder farmers will naturally consider new policies if they benefit their farming operations. The

results showed that researchers have policies and adaptations specifically for smallholder farmers; however, the average farmer will likely never read a journal publication. Researchers must develop communication strategies to assist farmers in accessing the latest research. Our suggestions include contacting local media, newspapers and radio stations, supplying pamphlets to suppliers of agricultural inputs, and developing websites or blogs. Once farmers become aware of the advice, they can access it by listening to local radio, reading local papers, or obtaining a helpful pamphlet. The next time they purchase inputs or go to a website, they will likely seek the advice researchers can offer them.

### 4.7.5. Policy Directed at the Research Community

The research gaps discussed in this review may interest the general research community. The research gaps identified in this review and other gaps are further discussed in the discussion section.

### 4.7.6. Policy Directed at Agribusiness

Goswami et al. [36] suggest that agricultural cooperatives, SHG, or existing federal programs extend credit to smallholder farmers. Magar et al. [4] suggest that agribusiness adapts and supports farmers during a pandemic.

### 4.7.7. Policy Directed at Technologists

We support Thulasiraman's [54] call for policy interventions to be used by farmers during pandemics.

## 5. Research Gaps

Researchers have identified the following research gaps. Asegie et al. [13] identified research gaps on the impact of COVID-19 that captures the seasonality and resilience capacity of households and on COVID-19's impact across the sex of household heads. Benedek et al. [47] identified a gap in the research concerning the clarification as to whether their findings on digital resources are generalisable.

Bloem and Farris [59] identified a gap in the research concerning COVID-19's impact on different socioeconomic groups. Ceballos et al. [42] identified a gap in the research concerning smallholder farming household vulnerability, nutritional food quality, and family food distribution. Dixon et al. [11] identified a gap in the research concerning the impacts on natural resources, whether adaptations to the impacts of COVID-19 lead to a boost in farm sustainability and diversification and finally, whether COVID-19 will be a significant influence in achieving a green economy. Goswami et al. [36] claim that the accurate assessment of the impact of dual crises on agricultural systems and accounting for the adaptive strategies is still beyond our knowledge and opens the scope for future research. Iese et al. [60] identified a gap in the research concerning food sovereignty and the impacts and adaptations to COVID-19.

Going forward, there is an urgent need to invest in technological interventions to be used in pandemic crises [54]. Meuwissen et al. [52] identified a gap in European resilient farming systems research. Han et al. [61] identified a gap in the research concerning the risk of viral transmission in food. Lang et al. [62] and Nayal et al. [46] identified a gap in the research concerning the application of new technologies to improve supply chain management. Van Hoyweghen et al. [63] and Varshney et al. [64] identified a gap in the research concerning the long-term impacts of COVID-19.

The critical research gaps that we have identified while developing this review include the following:

(a)  Research on the implications for smallholder farmers if a future pandemic is significantly more contagious and lethal than COVID-19.
(b)  Research on the first responder's activity for smallholder farmers when an unprecedented pandemic occurs.

(c)    The role of smallholder researchers and policymakers during the early stages of an unprecedented pandemic.

(d)    Continued research on how to mitigate the impacts of future pandemics.

(e)    The best ways to collect data during a pandemic crisis.

(f)    Continuing research on supply logistics for smallholder farmers.

(g)    Assessment methods to assess smallholders' readiness for future pandemics.

(h)    Controlling predatory pricing of essential food during pandemic crises.

(i)    Accessing labour for smallholder farms during a pandemic.

(j)    Production of homemade fertilisers.

(k)    Providing services for women, children, and the aged during a pandemic.

(l)    Overcoming language barriers during a pandemic.

(m)    The provision of essentials to smallholder farm households severely impacted by pandemic crises: Can we afford social security for vulnerable farm households? Cost–benefit analyses concerning the support of smallholder family households against the alternative?

(n)    Continued research concerning strategies to overcome barriers to communication for smallholder farmers in remote areas.

(o)    The provision of education to smallholder farmers.

(p)    The role of government concerning smallholder farm resilience for future pandemic crises.

(q)    The benefits of engaging farmers and agribusiness in policy development for future pandemic crises.

(r)    The real impacts of adaptations impact on smallholder farming during COVID-19.

## 6. Limitations

It has been challenging to conduct primary research on farms due to COVID-19 restrictions. However, despite the restrictions on movement, 33% of primary research studies used digital methods such as online surveys, phone surveys, and SMS.

Variation in the number of studies from low to middle to high-income countries and comparative studies suggests variation in research ethics in the countries. Forty-one studies were from low and middle-income nations, suggesting a more flexible approach to research in those regions; Only three studies from high-income countries suggest stricter research in rich countries during COVID-19. More global (comparative) studies, including studies from high-income counties, will enable more reliable comparisons and, possibly, additional knowledge of the continuum of adaptation options.

## 7. Conclusions

This review systematically analysed and documented the impacts, adaptations, barriers to adaptations, policies, and research gaps concerning COVID-19 and future pandemics. The impacts on smallholder farming were sometimes severe. The literature regarding smallholder farming discusses many valuable adaptations; however, the actual quantitative results of their implementation have yet to be included. Therefore, further research on the impact of adaptations is required. Barriers to implementing adaptations were considered in approximately 15% of the journal articles. The review documented many policy issues that will be invaluable to policymakers.

Research gaps were discussed in approximately 20% of the review articles. Research gaps were included in the discussion to be a priority for future pandemics. Broadly, the review has documented a substantial and extensive repository of knowledge that can be used to mitigate the impacts of future pandemics. This has also enabled discussion concerning various issues, and many suggestions for further research have been formed.

**Supplementary Materials:** The following supporting information can be downloaded at: https://www.mdpi.com/article/10.3390/land12020404/s1, Supplementary Table S1 and Table S2 include a table of the authors and works included in the systematic review and a table including policy and adaptation recommendations by authors. (In text citations are included).

**Author Contributions:** A.R.M. led the overall review, including the write-up of the draft; K.K.Z. and J.A.L. contributed to the design and method of the review, reading, editing, tweaking and proofreading the draft. All authors have read and agreed to the published version of the manuscript.

**Funding:** Charles Darwin University funded this research.

**Institutional Review Board Statement:** The research has been approved by the Charles Darwin University Ethics Committee.

**Conflicts of Interest:** The authors declare no conflict of interest.

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
