# Peer review of "Smallholder Farming during COVID-19: A Systematic Review Concerning Impacts, Adaptations, Barriers, Policy, and Planning for Future Pandemics"

_land, doi:10.3390/land12020404_

Round 1

Reviewer 1 Report (Previous Reviewer 1)

Thank you!

Author Response

Hi, it appears that you support the article for publication, however I don't understand why you found that the conclusions supported by the results was not applicable?

Reviewer 2 Report (Previous Reviewer 2)

In the section on research methodology (section 2), authors have said “We conducted a systematic literature review (SLR) using peer-reviewed literature published in four databases (Scopus, Web of Science, Agricola, PubMed) during 2020-2021”. Please include reviews of literature or studies of some years before 2020-2021 (say, 2015-2019). This will capture adaptation strategies and impact analysis before and during COVID-19.

Author Response

Hi, I spoke to my supervisor about 'including reviews of literature or studies of some years before 2020-2021 (say, 2015-2019). This will capture adaptation strategies and impact analysis before and during COVID-19.' We discussed this and she said that this was not really necessary. However the comment was also in the original resubmission so I researched the impacts of previous recent pandemics and I included this in the resubmission. I am quite willing to go back and find adaptations in the literature concerning recent pandemics. The manuscript has changed considerably after the first round of reviews. I have checked and rechecked the grammar and spelling multiple times and found no errors, but of course style can be subjective. Overall I am happy with your review however if I am to improve further I need to know specifics. I was very pleased to see that you did not mark anything as needing improvement.

Reviewer 3 Report (Previous Reviewer 4)

The authors have fixed the paper following the past some recommendations.

Author Response

Hi,

Thank you for recognising that errors have been addressed. I understand from your comments that you now support this manuscript.

This manuscript is a resubmission of an earlier submission. The following is a list of the peer review reports and author responses from that submission.

Round 1

Reviewer 1 Report

Dear authors,

thank you for this well designed literature review.

Despite the well structured methodological approach I find the results disappointingly short, simplified, and sometimes wrong. A mere list of policy recommendations for every single paper reviewed (pp. 18ff.) does not provide any insight or scientific value; let alone "help for policymakers" (l. 490). Analytical rigor is missing here.

The discussion is severely lacking substance. You present several statements that cannot be derived from your results. Other statements seem rather pointless to me.  E.g. your conclusion that "there needs to be more research focused on small-scale farming in many countries" (l. 462) is absolutely not backed by "no research output" in certain countries (Russia, Canada etc.) where small-scale farming plays a marginal role in the agri-food system. And that there is a general lack of research on small-scale farming and comparative studies with large-scale production is not deductable from your body of literature focused on COVID-19.

Also, you do not clearly differentiate between findings of reviewed studies and results of your own analysis (e.g. l. 537). This once more shows the lack of analytical focus.

There are many more examples that I will not go into detail here because from my personal and professional perspective the manuscript does not have the substance to be published.

Reviewer 2 Report

Dear Authors, thanks for writing a paper on this important topic. The idea of the paper is interesting and I have enjoyed reading the full paper. However, the paper quality will improve if it is revised before publication in the journal. While revising the paper, the author(s) should give attention to few comments and suggestions given below.

(1)    In section 2 (Research methodology), authors have said “We conducted a systematic literature review (SLR) using peer-reviewed literature published in four databases (Scopus, Web of Science, Agricola, PubMed) during 2020-2021”. If the literatures or studies of some years before 2020-2021 (say, during 2015-2019) were also included, then it would have been more interesting in terms of methodology for comparison. Authors could have captured adaptation strategies and impact analysis before and during COVID-19.

(2)     The summary points of policy recommendations given in Table 6 are nice. However, discussions given by authors should be improved by own analysis. Adaptation strategies may also be classified in relations to government/non-government policies, income/non-income related variables, technology/non-technology methods, climate/non-climate variables, small farmers vs big farmers etc.

(3)    While examining adaptation strategies and/or mitigation strategies of small farmers, it is important to know whether measures adopted by governments were successful or not. Some success stories may be highlighted for future policies.

(4)    Now, COVID-19 problems are almost over in many countries of the world. COVID-19 problems are not affecting adversely on small farming now. Therefore, writings in section 5 (limitations) and section 6(conclusions) should highlight this issue of no COVID-19 and way forward.

---x---

Reviewer 3 Report

The article presents information on small-scale farming vs COVID-19 which is an interesting subject. However, having carefully read and reflected on the content of the article, I note that: The article is more of a bibliographic review than a systematic review. This created discrepancy between what is expected in the systematic review vs what was presented in the paper. The methodology required enrichment, discussion requires clear criticism and reflection of literature used.  To be maintained as a systematic review, authors may consider improving their article. I suggest the following improvements:

Title

Line 2: Needs to be rephrased to make it informative. Its looks redundant in its current form.

Abstract

Line 15 16, 18: As much as including references in the abstract authenticate information, It’s highly discouraged. I suggest references to be deleted. Let the abstract be the authors reflection of the whole paper. The key results and recommendations not in context. This is because the results were general!

Introduction

The article provides seemingly useful information. However, it lacks a strong quantitative justification (crop, animal etc vs covid19 vs livelihood etc) that justifies the listed study objectives. No evidence on specific policies and how they worked; so one wonders how objective three (policy) was arrived at. It’s also good to note that smallholder farmers may or may not own/control land or production facilities. Is there any evidence that smallholder famers who rent land or production facilities were affected?. In nutshell, the introduction lacks in-depth state of art information on the variables under consideration.

Line 44: “This review found……..” which review. The way this is being reported isn’t clear; as if results…. This paragraph doesn’t not logically connect with the preceding paragraph.

Line 46-48: The two sentences should be recasted and presented as one concise sentence.

Line 83: Synthesize not Synthesise

There are few grammatical and spelling errors, consider proof reading

Methodology:

Line 89-90: from we…., delete this sentence because its repeated from the subsequent sections it’s supposed to be

Figure 2 and 3: Nvivo is already described in the text and I feel that’s adequate. So.  I suggest Figure 2 and 3 to be deleted because they seem not to add value to the work.

Figure caption consistency: In figure 3 the caption is above and below in figure 4… The authors need to stick to one format as required by the journal.

Table 1: Could be modified; to include country/region, type of article (if document different types were used), objective of the study etc to make it more relevant. I suggest that you place those articles into clusters like policy, adaptation, etc depending on where their major results are.

Search strategy: A part from limiting with English language. Document type included in this study is not mentioned (search by document type needs to be clarified). Also, the search key words covid-19 OR corona OR covid19* AND smallholder* OR small-scale OR subsistence OR peasant were used. Including disaggregated components such as agriculture, crop, livestock etc could improve the quality of search. Besides, in the exclusion criteria, its clearly stated that articles not referring to crop or livestock farming were excluded!

Selection phase: The three broad exclusion criteria were used for overall document assessment. I miss the point on how exactly the articles were reviewed in detail. Leave alone use of Nvivo! Line 102: ……. studies not on small-scale farming except in small-scale and large-scale cases. This is not clear”. Line 103 does not refer to crop or livestock farming”……. Why were this words not part of the search strategy?

Analysis: Authors used Nvivo to analyse article texts. This is sufficient to provide evidence from “a bibliographic review point of view”. However, deeper systematic review (aim of this paper) required authors to individually review the 53 articles in depth for understand the nature of results, discussion and arguments presented. This way, maybe logical reflections and output could have been revealed.

Results

Authors should use open tables.

Most results are presented as if the work was a bibliographic review of the sort. The content presented lacks in-depth analysis.

Fig 4: It’s not easy to determine from the lower side of the scale, location of studies.

Line 268-273; COVID19 effects are stated in a descriptive approach. At the end, there is this statement “with these quantitative results……….”. I find it strange because I did not see any valid quantitative evidence. Authors need to revisit those article and extract the quantitative evidence that they can substantiate.

Supply, labour and inputs; The effects are summarised. But nothing new seems to be added. Infact, the results are almost the same effects reported in the background. There are no reflections, and gaps identified. Besides, Literature reported is not even critiqued. This made the review to lack a well-articulated output.

Social aspects: Effect of COVID19 on men, women and children are summarised. The authors point out the effects on farming and food supply. However, this section has parts which are not logically presented (two lines, three lines etc. These should be merged in a logical way). Line 343-344 points out the effect of COVID19 on nutrition status of women, but the preceding sentence doesn’t discern meaning! This section needs to rewritten in a logical way. Also, authors state that research gaps exist here and there without clear basis!

Discussion and gaps

Line 445-461: Authors list a lot of recommendations in one paragraph. Then in the next paragraph, authors try to validate a few of the recommendations. Reading this part wasn’t compelling at all as it was tenuous and superficial! Authors should focus on key results emanating from the review and put them into context.

Conclusion

Line 525-533: Key findings are listed but not put to context.

Reviewer 4 Report

The topic is of potential interest to readers.  In its present form the quality of the work and the outcome are difficult to judge.

The paper needs to be improved in content and structure, with much clearer signposting of objective, methods, results and discussion, with focused conclusions.

There is a very loose link between the theoretical foundation and food security in rural areas. A clear storyline is lacking. The authors can use the following papers to improve link between the theoretical foundation and food security in the Introduction.

https://doi.org/10.1080/00036846.2022.2119199

https://doi.org/10.1080/09709274.2017.1317504

https://doi.org/10.1080/10371656.2021.1895471

Table 1, Figure 2, and 3 are redundant.

The section of “Discussion and Research Gaps” should be changed to the “Discussion”.

The section of “Limitations” should be moved after the “Conclusion”.

The conclusion section need to rewritten, also, more clearly and more linked you the main insights from the research. At the beginning of the Conclusion Section, a paragraph highlighting the main contribution and a brief description of the article is missing.